# Role of A-Kinase Anchoring Protein 1 in Retinal Ganglion Cells: Neurodegeneration and Neuroprotection

**DOI:** 10.3390/cells12111539

**Published:** 2023-06-03

**Authors:** Tonking Bastola, Guy A. Perkins, Keun-Young Kim, Seunghwan Choi, Jin-Woo Kwon, Ziyao Shen, Stefan Strack, Won-Kyu Ju

**Affiliations:** 1Hamilton Glaucoma Center and Shiley Eye Institute, The Viterbi Family Department of Ophthalmology, University of California San Diego, La Jolla, CA 92093, USA; tbastola@health.ucsd.edu (T.B.); sec009@health.ucsd.edu (S.C.); j8kwon@health.ucsd.edu (J.-W.K.); zis003@ucsd.edu (Z.S.); 2National Center for Microscopy and Imaging Research, Department of Neurosciences, University of California San Diego, La Jolla, CA 92093, USA; guyaperkins@gmail.com (G.A.P.); kkim@health.ucsd.edu (K.-Y.K.); 3Department of Ophthalmology and Visual Science, College of Medicine, The Catholic University of Korea, Seoul 06591, Republic of Korea; 4Department of Ophthalmology and Visual Science, St. Vincent’s Hospital, Jungbu-daero 93, Paldal-gu, Suwon 16247, Republic of Korea; 5Department of Pharmacology, Iowa Neuroscience Institute, University of Iowa, Iowa City, IA 52242, USA; stefan-strack@uiowa.edu

**Keywords:** AKAP1, DRP1, mitochondrial dynamics, mitochondrial fission, glaucoma, retinal ganglion cell, neurodegeneration, neuroprotection

## Abstract

A-Kinase anchoring protein 1 (AKAP1) is a multifunctional mitochondrial scaffold protein that regulates mitochondrial dynamics, bioenergetics, and calcium homeostasis by anchoring several proteins, including protein kinase A, to the outer mitochondrial membrane. Glaucoma is a complex, multifactorial disease characterized by a slow and progressive degeneration of the optic nerve and retinal ganglion cells (RGCs), ultimately resulting in vision loss. Impairment of the mitochondrial network and function is linked to glaucomatous neurodegeneration. Loss of AKAP1 induces dynamin-related protein 1 dephosphorylation-mediated mitochondrial fragmentation and loss of RGCs. Elevated intraocular pressure triggers a significant reduction in AKAP1 protein expression in the glaucomatous retina. Amplification of AKAP1 expression protects RGCs from oxidative stress. Hence, modulation of AKAP1 could be considered a potential therapeutic target for neuroprotective intervention in glaucoma and other mitochondria-associated optic neuropathies. This review covers the current research on the role of AKAP1 in the maintenance of mitochondrial dynamics, bioenergetics, and mitophagy in RGCs and provides a scientific basis to identify and develop new therapeutic strategies that could protect RGCs and their axons in glaucoma.

## 1. Introduction

### 1.1. AKAP1

A-Kinase anchoring protein 1 (AKAP1) is a multifunctional mitochondrial scaffold protein that anchors protein kinase A (PKA) and other signaling proteins such as protein phosphate 1 (PP1), Ca^2+^-dependent phosphatase calcineurin (CaN), phosphodiesterase 4 (PDE4), and Src kinase to the outer mitochondrial membrane (OMM) [1,2,3]. As an OMM-targeted AKAP, AKAP1 regulates mitochondrial dynamics and contributes to the maintenance of mitochondrial networks and bioenergetics, cellular calcium homeostasis, and viability [4]. AKAP1 possesses a characteristic PKA binding domain capable of binding both regulatory subunits (type-I and type-II) of PKA. Thus, it is also known as D-AKAP1 (dual specificity AKAP1). Other common names are AKAP121 in mice and rats, AKAP149 in humans, and S-AKAP84, which is a unique isoform found in sperm [5,6,7]. Multiple AKAP1 splice variants are expressed from the *AKAP1* gene in a variety of tissues, including the brain, heart, liver, kidney, and skeletal muscle [8]. The full-length encoded AKAP1 protein (Figure 1) consists of an N-terminal OMM domain with a mitochondrial targeting (MT) sequence (1-30aa), a PKA binding site, a highly conserved c-terminal RNA binding K homology (KH) domain, and a Tudor domain [9,10,11,12]. The MT motif is present in the first 30 residues of all AKAP1 isoforms and consists of a hydrophobic helix followed by positively charged amino acids. It is necessary for attaching the AKAP1 scaffold to the OMM [13,14,15]. The PKA binding domain, which interacts with the dimerization/docking (D/D) domain of the PKA regulatory subunit, is a 14–18 amino acid fragment that is highly conserved in structure but is poorly conserved across the AKAP family [13]. Some AKAP1 splice variants (e.g., S-AKAP84) lack the C-terminal KH and Tudor domains [10,11]. Interactions between the KH domain and AU-rich RNAs or single-strand DNA have been observed [9]. Within the KH domain, an intact hydrophobic groove is essential for proper RNA binding [14]. The Tudor domain of AKAP1 is made up of five antiparallel beta-sheets that are folded into a structure resembling a barrel. This structure can detect symmetrically di-methylated arginine and possesses mild RNase activity. As a result, the KH–Tudor domain is responsible for regulating translation as well as the stability and subcellular localization of binding RNAs [14,15]. The tubulin-binding motif, which is found inside the N-terminal MT motif (1–30 aa), is another characteristic shared by AKAP149, AKAP121, and S-AKAP84. The conserved area known as the tubulin-binding motif interacts with both tubulin polymers and soluble microtubules [9].

AKAP1 integrates various intercellular signal transduction cascades via interaction with proteins and mRNAs at the OMM to maintain mitochondrial homeostasis [3]. Thus, AKAP1 is considered a mitochondrial signaling hub and plays vital roles in the regulation of mitochondrial function and structure. AKAP1 regulates mitochondrial structure and function by recruiting PKA to the OMM. The cyclic adenosine 3′5′-monophosphate (cAMP)/PKA signaling pathway is one of the cell’s signaling cascades that most affects mitochondria. By anchoring PKA and modulating dynamin-related protein 1 (DRP1) phosphorylation, AKAP1 promotes mitochondrial elongation and increases cell survival, demonstrating its pivotal function in the PKA/DRP1 signaling axis [2]. Furthermore, AKAP1 could mediate mitochondrial bioenergetics by augmenting ATP synthesis and the mitochondrial membrane potential [16,17]. In the CNS, AKAP1 prevents neuronal cell loss during a cerebral ischemic stroke by preserving the activity of the respiratory chain, mitigating DRP1-mediated mitochondrial fission and superoxide production, and delaying the dysregulation of Ca^2+^ [18]. AKAP1/PKA axis-mediated neuroprotective mitochondrial remodeling protected HT22 cells by phosphorylating DRP1 and blocking glutamate-induced oxidative stress [19]. In addition, augmenting AKAP1 expression improved dendritic growth and reduced synaptic density in cultured hippocampal neurons [20].

Recent evidence indicates that AKAP-mediated signaling is critical for several fundamental neurological processes and suggests that understanding the function of AKAPs in nervous system disorders may develop novel therapeutic approaches [21]. AKAPs are involved in a wide range of neuronal functions, including neurotransmitter release [22], synaptic plasticity [23], neuronal excitability [24], ion channel function, and gene expression [25,26]. They are also crucial for adequately developing and maintaining neuronal structures such as dendrites and axons [16].

### 1.2. cAMP/PKA Signaling and AKAP1 Complex

cAMP, a ubiquitous second messenger, regulates growth, differentiation, and cell survival and contributes to the signal transduction pathway activated by growth factors, hormones, and neurotransmitters [21]. It is synthesized from the catalytic cyclization of ATP by adenylyl cyclases (ACs) [22] and acts through three types of effector proteins: PKA, exchange proteins activated by cAMP (EPACs), and cyclic nucleotide-gated ion channels (CNGs/HCNs) [23]. PKA is the major sensor of cAMP and could selectively phosphorylate substrates in response to different stimuli. The localization of PKA is guided by AKAPs that target it to specific substrates [24]. AKAP1 attaches PKA type-I and type-II regulatory subunits to the OMM in response to cAMP-signaling upon PKA activation [24]. AKAP1 also tethers the cyclic nucleotide PDE4, which hydrolyzes cAMP to inhibit cAMP-signal transduction [12]. Since AKAPs interact with the regulatory subunits rather than the catalytic subunits of PKA, localized signaling within the cell still needs the cAMP-mediated release of the active catalytic subunit of PKA in order to phosphorylate substrates and achieve compartmentalized signaling [20]. AKAPs provide the architectural infrastructure for the specialization of the cAMP signaling network by bringing together unique combinations of upstream and downstream signaling molecules. AKAP1/PKA signaling is crucial for adequately coordinating dendritic outgrowth during neuronal development. Dendritic complexity is increased when AKAP1 is overexpressed during development, but the number of synapses is not affected [16]. Therefore, the role of AKAP1 in coordinating cAMP/PKA signaling at the OMM is critical to maintaining neuronal health, development, metabolism, and physiology.

### 1.3. AKAP1 and Glaucomatous Retinal Ganglion Cells

Retinal ganglion cells (RGCs) are solely responsible for transmitting visual information from the retina of the eye to the visual cortex of the brain. In humans, approximately 50% of RGC axons cross over to the opposite side at the optic chiasm, and most of these axons end up in a specific area of the brain called the lateral geniculate nucleus. However, in rodents, almost all RGC axons cross over at the optic chiasm, and some of the axons form synapses in the superior colliculus [25,26,27,28]. RGC somas are located in the innermost layer of the retina, called the ganglion cell layer, and their dendrites ramify in the inner plexiform layer, where they receive signals from bipolar cell axons and amacrine cell processes. After processing this information, RGCs transmit it to the brain through their axons, which travel along the nerve fiber layer and optic nerve [29]. There are several types of RGCs, each with different properties and functions. Some RGCs are sensitive to light intensity and color, while others respond to movement or changes in the visual field. RGCs also vary in spatial resolution, with some cells able to detect fine details and others more suited to detecting larger objects [30]. In addition to their role in vision, RGCs also play an important role in other physiological processes, such as regulating circadian rhythms and modulating pupil size [31]. In glaucoma, RGCs and their axon degeneration are the main characteristics, resulting in vision loss. Until now, there have been several risk factors in glaucoma pathophysiology, and these include elevated intraocular pressure (IOP) [32,33], oxidative stress [34], autoimmune response [35,36], glial cell activation [37,38], mitochondrial dysfunction [3], excitotoxicity [39], and aging [40]. However, the precise pathophysiological mechanisms underlying the RGC soma and its axon degeneration still need to be elucidated. Evidence from our group demonstrated that mitochondrial dysfunction plays a critical role in the degeneration of the optic nerve axon and progressive RGC death in glaucomatous neurodegeneration [3,41,42]. Since AKAP1 protects neuronal cells and promotes neurite outgrowth in the central nervous system [18], our recent studies indicate that glaucomatous insults, including elevated IOP and oxidative stress, induce AKAP1 deficiency in RGCs [43], and increasing AKAP1 expression promotes RGC survival against oxidative stress [3]. Thus, these findings suggest that AKAP1 may play an important role in mitochondrial protection in RGCs against glaucomatous neurodegeneration.

## 2. AKAP1 and Its Role in Mitochondria

AKAP1 is a primary regulatory molecule in various mitochondrial activities such as oxidative phosphorylation, mitochondrial membrane potential, Ca^2+^ homeostasis, apoptosis, and the phosphorylation of various mitochondrial respiratory chain substrate molecules [44]. Recent evidence indicates that AKAP1 impacts the regulation of mitochondrial dynamics, bioenergetics, and mitophagy [18,43,45] (Figure 2).

### 2.1. AKAP1 and Mitochondrial Dynamics

Mitochondria undergo four dynamic processes: fusion, fission, motility, and mitophagy. Mitochondrial fusion involves the joining of distinct mitochondria to form new mitochondria, while fission involves the division of a mitochondrion into separate mitochondria. Mitochondria are transported to specific sites within the cell by the microtubule network, a process called mitochondrial motility. Nonfunctional mitochondria are selectively eliminated through the mitophagy process, which involves machinery like that involved in autophagy [46]. Proper balance among these dynamic processes is critical for cellular homeostasis, and alteration in any of these processes has been linked to various pathological conditions. Multilayered regulatory mechanisms and signaling molecules ensure the proper tuning of these dynamic processes [47,48].

Mitochondria undergo continuous dynamics, fission and fusion, in a highly coordinated, balanced fashion based on the particular requirements for cellular functions such as survival, growth, division, and distribution [49]. Mitochondrial fission is mediated by DRP1 and Fis1, localized at the OMM, while mitochondrial fusion is mediated by optic atrophy type 1 (OPA1) and mitofusin (MFN) 1 and 2, localized in the inner mitochondrial membrane and the OMM, respectively. AKAP1 contributes to mitochondrial dynamics, fission, and fusion and regulates mitochondrial fission by modulating DRP1 activity through PKA. Dephosphorylation of DRP1 at Serine 637 (S637 in human isoform 1 and S656 in rat isoform 1) by CaN promotes mitochondrial fission, while PKA-mediated phosphorylation of DRP1 S637 promotes mitochondrial fusion by lowering DRP1 fission activity [50,51,52]. On the other hand, DRP1 phosphorylation at Serine 616 (S616) promotes mitochondrial fission, which is governed by cyclin-dependent kinase 1/cyclin B [53,54].

In addition, Dickey and Strack showed that PKA/AKAP1-mediated phosphorylation of DRP1 S637 increased mitochondrial length in cultured rat hippocampal neurons, leading to enhancement of dendritic outgrowth and reduction of synapse number [16]. A recent study from our group showed elevated IOP-induced AKAP1 loss, activated CaN, and increased dephosphorylation of DRP1 S637 in RGCs of glaucomatous DBA/2J mice [43]. Furthermore, AKAP1 deficiency led to increases in CaN and total DRP1 protein expression and an increase in dephosphorylation of DRP1 S637 in the retina of AKAP1 knockout (^−/−^) mice [43]. Of interest, electron microscopy (EM) analysis revealed that AKAP1 deficiency caused mitochondrial fragmentation and loss in *AKAP1*^−/−^ RGCs [43]. Thus, these findings strongly support the notion that AKAP1 plays a critical role in regulating mitochondrial dynamics in glaucomatous RGCs.

### 2.2. AKAP1 and Mitochondrial Bioenergetics

Mitochondria provide energy to mammalian cells through the mitochondrial oxidative phosphorylation (OXPHOS) system. This system consists of five protein complexes and two electron carriers [55]. PKA regulates mitochondrial respiration by phosphorylating substrates like NDUFS4 and cytochrome c oxidase (COX) [56]. PKA-mediated phosphorylation of COX promotes allosteric ATP inhibition, while dephosphorylation by Ca^2+^ enhances respiration and ATP production [57,58]. It is unclear whether PKA acts at the IMM through holoenzymes or AKAP anchoring.

At the OMM, AKAP1 also anchors the Src tyrosine kinase via the tyrosine phosphatase PTPD1. Interestingly, phosphorylation of COX at a tyrosine residue by mitochondrial Src promotes ATP synthesis [59,60], but again, the connection between OMM and IMM or matrix localized Src has yet to be elucidated. Impairment of OXPHOS, mitochondrial respiration, and ATP production systems induces oxidative stress by generating excessive reactive oxygen species (ROS) and metabolic stress, ultimately causing mitochondrial dysfunction and apoptosis in various mitochondrial diseases [61]. Flippo and coworkers demonstrated that AKAP1 plays a critical role in maintaining the activity of respiratory complex II, an essential component of OXPHOS that generates ATP in mitochondria. In addition, deletion of AKAP1 dysregulates OXPHOS complex II, increases superoxide production, and impairs Ca^2+^ homeostasis in hippocampal neurons in response to glutamate excitotoxicity. These results suggest that AKAP1 is a key regulator of mitochondrial bioenergetics [18]. Although AKAP1 is located at the OMM, PKA and cAMP are located inside the mitochondria as well and thus can act on cristae proteins [62]. Interestingly, our recent study also showed that AKAP1 deficiency dysregulated OXPHOS complexes by increasing the level of Complex II and decreasing the level of Complex III-V in the retina of *AKAP1*^−/−^ mice [43].

Based on these findings and increased mitochondrial fission in *AKAP1^−/−^* RGCs, we questioned, “Could AKAP1 deficiency trigger an energy deficit via a reduction in the rate of ATP production in RGCs?” Utilizing the modeling framework proposed by Song and coworkers [63], we found that the crista density was significantly lower in *AKAP1*^−/−^ RGC mitochondria, corresponding to a lower rate of ATP production per mitochondrion, being about two-thirds of WT mitochondria [3]. In addition, we observed that the rate of ATP production per unit mitochondrial volume was reduced by over two-fold in *AKAP1^−/−^* RGCs [3]. This significant energy deficit in the *AKAP1*^−/−^ RGC was due to a decreased rate of ATP production per mitochondrion and a reduced volume density of mitochondria. These results suggest that AKAP1 deficiency-mediated reductions in ATP production contribute to functional impairment and the loss of RGCs.

### 2.3. AKAP1 and Mitophagy

Mitochondrial dysfunction would be harmful to cells, but autophagy and mitophagy processes maintain energy metabolism in the case of such dysfunction [64]. Mitophagy is the conserved cellular process that removes damaged mitochondria to preserve healthy ones [65] and can occur selectively or non-selectively [66]. In yeast, mitophagy is selective [65,67], whereas non-selective mitophagy happens during nutrient deprivation when mitochondria are present in autophagosomes alongside other cellular components [68,69]. In most cases, mitophagy is associated with the removal of damaged mitochondria, but it has also been observed to play a role in reticulocyte differentiation and lens maturation. i.e., when mitochondria are no longer needed [64]. Inhibition of RGC differentiation following pharmacological or genetic inhibition of mitophagy confirmed that programmed mitophagy is required for the differentiation of RGC during embryogenesis in mice [70]. However, failure of properly functioning mitophagy can result in the accumulation of damaged mitochondria characterized by altered cristae structure, dysfunction in bioenergetics, and subsequent cellular dysfunction, which ultimately leads to tissue and organ damage [71,72]. Extensive research has been conducted on two major mitophagy pathways in mammals, namely the receptor-dependent pathway and the ubiquitin-mediated pathway. The ubiquitin-mediated pathway involved in mitochondrial quality control is regulated by PTEN-induced kinase I (PINK1) and Parkin (PARK2)-mediated mitophagy [73,74], whereas the receptor-dependent mitophagy pathway includes the involvement of Nip3-like protein X (NIX) and Bcl-2 19-kDa interacting protein 3 (BNIP3). Additionally, autophagy and beclin 1 regulator 1 (AM-BRA1) function separately from PARKIN/P62 and FK506 binding protein 8 (FKBP8), which interact with LC3. Another protein involved in this pathway is FUN14 domain-containing protein 1 (FUNDC1) [75,76,77].

AKAP1 can regulate mitophagy by interacting with PINK1, which is localized to the OMM and is activated in response to mitochondrial damage that initiates mitophagy [78]. PINK1-mediated regulation of the PKA/AKAP1 signaling axis selectively increases DRP1-mediated fission in damaged mitochondria, hence guaranteeing mitophagy initiation in mammalian cells [79]. Recent evidence demonstrated that AKAP1 deficiency-induced autophagosome/mitophagosome formation in RGCs by elevating LC3-II expression and decreasing SQSTM1/p62 expression in *AKAP1*^−/−^ mice [43]. Similarly, a study on the critical function of AKAP1 in cardiac responses to myocardial ischemia showed that *Akap1* deletion caused mitochondrial abnormalities and increased cardiac mitophagy. The autophagy inhibitor 3-methyladenine prevented further decline in cardiac function in *AKAP1*^−/−^ mice [45]. These findings suggest that AKAP1 deficiency may compromise mitophagy and, in turn, trigger mitochondrial dysfunction. Thus, AKAP1 could be crucial in maintaining mitochondrial health by promoting mitophagy.

## 3. AKAP1 in Neurodegeneration

### 3.1. AKAP1 in Glaucomatous Neurodegeneration

Glaucoma is a multifactorial neurodegenerative disease that causes slow, progressive, and irreversible degeneration of RGCs and deterioration of their axons. By 2040, the global prevalence of glaucoma is projected to rise to approximately 111.8 million individuals. Glaucoma progression can be examined by characteristic visual field defects, optic disc changes, and thinning of the retinal nerve fiber layer or macular RGC-inner plexiform layer [80]. Accumulating evidence indicates that multiple molecular and biological pathways, including oxidative stress and mitochondrial dysfunction, are involved in glaucoma pathogenesis [81]. Despite the high prevalence of the disease, the factors contributing to glaucoma progression are currently not well characterized. The IOP is the only proven modifiable and treatable risk factor. However, lowering the IOP is often not enough to prevent disease progression, leading to RGC death.

Recent evidence from our group and others demonstrated that impaired mitochondrial dynamics, metabolic stress, and mitochondrial dysfunction triggered by glaucomatous risk factors were crucial to RGC death in glaucomatous neurodegeneration [42,82,83,84,85,86,87,88,89]. Despite the widely appreciated disease relevance of mitochondrial dysfunction and loss, the molecular and cellular mechanisms underlying compromised mitochondrial structure and function and RGC degeneration in glaucoma need to be better understood. In addition, our recent studies demonstrated the critical contribution of mitochondrial dysfunction, oxidative stress, and apoptotic cell death in RGCs in experimental models of glaucoma [84,90]. Of note, we found that glaucomatous RGCs showed a significant loss of AKAP1, leading to the activation of CaN, an increase in DRP1 S637 dephosphorylation, and the induction of mitochondrial fragmentation and loss [43]. Emerging evidence from serial block-face scanning EM and EM tomographic analyses indicated a rise in abnormal elongation of mitochondria and an intensified pattern of branched mitochondria in the RGCs of glaucomatous DBA/2J mice [3]. In light of these findings, which align with our previous studies revealing fragmented mitochondria in glaucomatous RGC somas and axons [41], our findings suggest that the observed abnormal elongation and branching of mitochondria may have a potential correlation with the early or intermediate progression of RGC damage in glaucoma, after axonal degeneration. Of interest, these atypical mitochondrial phenotypes in RGCs suggest a considerable energy demand necessary for their survival against glaucomatous insults. These findings suggest the possibility that AKAP1-mediated phosphorylation of DRP1 S637 protects RGCs by promoting mitochondrial fusion activity against glaucomatous neurodegeneration.

### 3.2. AKAP1 in Other Neurodegenerative Diseases

Neurodegenerative diseases are distinguished by a progressive depletion of neuronal function and a slow, steady deterioration of selectively susceptible neurons as the disease progresses. Mitochondrial homeostasis is disrupted in several neurodegenerative diseases such as Alzheimer’s (AD) [91,92], Parkinson’s (PD) [93], and Huntington’s disease [94], as well as amyotrophic lateral sclerosis [95]. AKAP1/PKA signaling stimulates synaptic plasticity, neuronal growth, and dopamine production in neurons [16,96,97]. A reduction in AKAP1/PKA signaling contributes to early-stage degeneration in various neurodegenerative disorders, including AD and PD [96], suggesting a potential role of PKA/AKAP1 signaling in neurodegeneration and neuroprotection. Epistasis experiments using a DRP1 mutant with a deleted phosphorylation site suggested that cAMP and PKA/AKAP1 enhance mitochondrial elongation and neuronal survival by inhibiting DRP1 through a conserved PKA site [2,16].

Using a transient middle cerebral artery occlusion model of focal ischemia, Flippo and coworkers reported that AKAP1-deficient male mice were more vulnerable to focal brain ischemia and had smaller mitochondria yet a greater number of mitochondria and endoplasmic reticulum (ER) connections [18]. In addition, AKAP1 deletion dysregulated electron transport chain complex II, increased superoxide levels, and altered Ca^2+^ homeostasis in glutamate-exposed neurons [18]. Furthermore, neurons lacking AKAP1 showed reduced DRP1 S637 phosphorylation and experienced Ca^2+^ dysregulation [18].This study suggests that AKAP1/PKA inhibits DRP1-dependent mitochondrial fission, protecting neurons from ischemic stroke by sustaining respiratory chain activity, reducing superoxide production, and delaying Ca^2+^ dysregulation [18]. Similarly, a recent study demonstrated that endogenous AKAP1 was reduced in primary neurons treated with Aβ_42_ peptide in vitro and in the hippocampus and cortex of asymptomatic and symptomatic AD animals in vivo [98]. In the study, transiently expressing wild-type AKAP1 but not the PKA-binding deficient mutant of AKAP1 reduced mitochondrial fission, dendritic retraction, and death in Aβ_42_-treated primary neurons [98]. In addition, loss of AKAP1/PKA contributed to mitochondrial pathology and neurodegeneration in an in vitro model of AD [98].

## 4. AKAP1 in Neuroprotection

AKAP1 protects neuronal cells from death by inhibiting DRP1-dependent mitochondrial fission in ischemic brain injury, while AKAP1 deficiency results in DRP1 dephosphorylation, OXPHOS complex II dysfunctions, elevated ROS production, and disruption of calcium balance in neurons exposed to glutamate-induced toxicity [18]. Furthermore, upregulating AKAP1 expression restored the transport of mitochondria in the dendrites of cortical neurons lacking PINK1, utilizing the transport protein Miro-2 [99]. These findings suggest that the AKAP1/PKA axis inhibits mitochondrial fission or fragmentation, which is critical to protecting against neuronal cell death.

AD is a prevalent neurodegenerative disease potentially linked to mitochondrial dysfunction [92,100,101,102,103,104]. Mechanisms underlying mitochondrial dysfunction in AD include abnormal mitochondrial dynamics, deficits in mitochondrial trafficking and distribution, impaired mitochondrial biogenesis, abnormal ER-mitochondrial interaction, and impaired mitophagy. Mitochondrial dysfunction is an early and prominent feature of AD, suggesting a critical role for mitochondria in the early stages of AD pathogenesis [105,106]. Importantly, these pathological phenotypes of mitochondrial dysfunction in the brain of AD are like those in the glaucomatous retina, but the potential links and mechanisms underlying mitochondrial dysfunction in the retina of AD pathogenesis are not well understood. In AD mouse models and amyloid beta (Aβ) accumulation scenarios, impaired mitochondrial dynamics play a role in neurodegeneration [92,104,107]. Aβ oligomers have been found to exert inhibitory effects on the PKA/CREB pathway [108], and stimulating PKA in hippocampal neurons can mitigate the impact of acute Aβ expression on mitochondrial trafficking and the initiation of cell death [109]. A previous study showed that an agonist for estrogen receptor β (ERβ) that interacts with AKAP1 and is specifically positioned within mitochondria triggered mitochondrial fusion by increasing DRP1 S637 phosphorylation, and this led to neuroprotection in hippocampal neurons against Aβ accumulation [110]. Using an in vitro culture system, a recent study suggests that the enhancement of endogenous AKAP1 levels is protective against Aβ-induced mitochondrial pathology and neurodegeneration [98].

Using an RGC culture system, emerging evidence from our group demonstrated that paraquat (PQ)-induced oxidative stress significantly decreased mitochondrial respiration in RGCs [3]. In parallel, oxidative stress produced increased apoptosis in RGCs. Intriguingly, overexpression of AKAP1 by in vitro transduction of adeno-associated virus type 2 (AAV2)-AKAP1 resulted in a substantial enhancement of mitochondrial activity and cell viability. In addition, AKAP1 expression significantly prevented apoptotic cell death in RGCs [3]. Considering our research findings, we suggest that amplifying AKAP1 expression through an in vivo AAV delivery system holds promise as a therapeutic approach to protecting the integrity and function of the mitochondrial network. This, in turn, can enhance the survival of RGCs and provide resilience against glaucomatous insults, including elevated IOP and oxidative stress. Thus, we proposed that sustained AKAP1 expression can inhibit CaN activation and promote DRP1 phosphorylation at the S637 site. This, in turn, can prevent mitochondrial fission or induce mitochondrial elongation and preserve mitochondrial function, leading to RGC survival (Figure 3).

## 5. Conclusions and Future Perspectives

The disruption of mitochondrial dynamics is correlated with mitochondrial bioenergetic dysfunction, leading to neuronal loss in many neurodegenerative diseases, including glaucoma [41,42,82]. Increasing evidence indicates that impaired mitochondrial dynamics is associated with damage to the RGC and its axon in glaucomatous neurodegeneration [41,42,82,86]. Our findings and others have improved the understanding of elevated IOP- and/or oxidative stress-mediated mitochondrial dysfunction in glaucoma, demonstrating that elevated IOP and/or oxidative stress induce excessive mitochondrial fragmentation and dysfunction in glaucomatous RGCs. We also uncovered, importantly, that the modulation of mitochondrial dynamics by inhibiting DRP1 activity rescued RGCs and their axons by preserving mitochondrial integrity in glaucomatous DBA/2J mice [41]. These findings call for an increased understanding of the global picture of mitochondrial signaling, dynamics, and bioenergetics and further testing of the functional and structural endpoints impacting the survival of RGCs in glaucomatous neurodegeneration. AKAP1 plays a key role in RGC survival by facilitating DRP1 phosphorylation and subsequent mitochondrial elongation, increasing ATP generation, and maintaining proper mitochondrial membrane potential [16,17], resulting in neuroprotection [2].

Here, we have discussed the role of AKAP1 in mitochondrial protein interactions, signaling, dynamics, and bioenergetics in RGC degeneration and protection. Future studies could address the following points: (1) How does mitochondrial dynamics modulated by AKAP1 influence bioenergetics and susceptibility in models of glaucoma not covered in this review? (2) Do elongated mitochondria sustain higher metabolic activity? [111] In HeLa cells, would unopposed mitochondrial fusion due to PKA-mediated DRP1 phosphorylation be protective in glaucoma models? PKA-mediated DRP1 phosphorylation increased oxygen consumption and ATP production, and spared mitochondria from autophagic degradation [17]. However, increased oxygen consumption could also lead to increased ROS, and this might be deleterious despite the increase in ATP production. Does a similar mechanism operate in the RGC and its axon? (3) What is the relationship between AKAP1 signaling and mitochondrial structure? (4) Does AKAP1-mediated mitochondrial neuroprotection in RGCs extend to the preservation of the central visual pathway? Since the amplification of AKAP1 gene expression by in vitro transduction of the AAV system is neuroprotective in cultured RGCs against oxidative stress [3], we will further validate the protective effect of AKAP1 on mitochondrial dynamics and bioenergetics in RGCs and their axons in experimental models of glaucoma using AAV-mediated gene therapeutic approaches. Collectively, these proposed studies may help to identify the role of AKAP1 as a potential therapeutic target and open new innovative strategies for mitochondrial dynamics-related neuroprotective intervention in glaucoma and other optic neuropathies.

## Figures and Tables

**Figure 1 cells-12-01539-f001:**
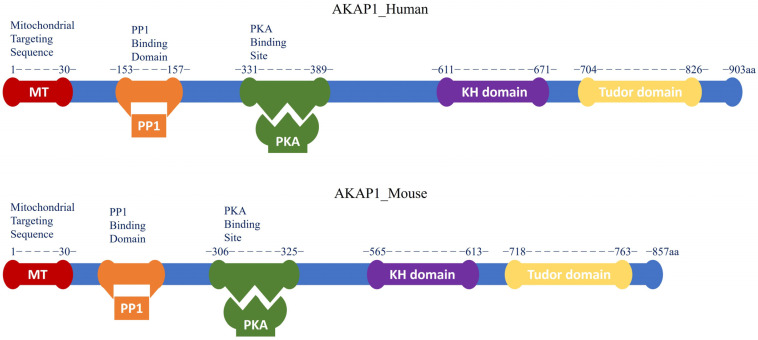
Schematic diagram of full-length AKAP1 protein in humans and mice. The different color boxes represent the different protein domains of full-length AKAP1 (Blue), indicated as N-terminal MT domain (Red), PP1 binding domain (Orange), PKA binding domain (Green), C-terminal mRNA binding KH domain (Purple) and Tudor domain (Yellow). Its domain structure sheds light on the molecular mechanisms by which AKAP1 modulates many cellular signaling pathways.

**Figure 2 cells-12-01539-f002:**
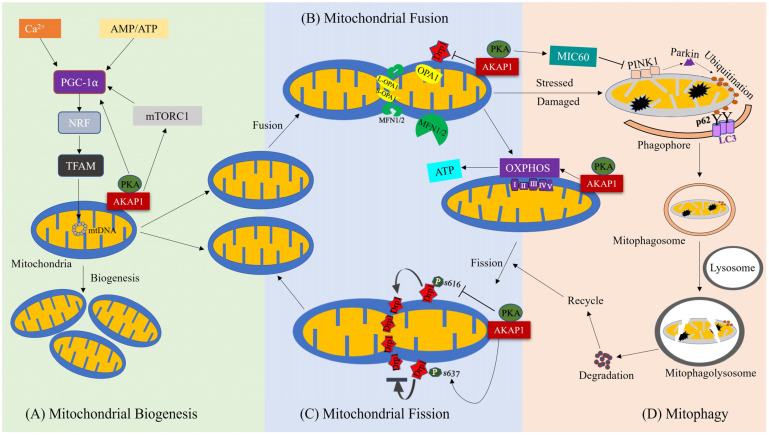
Role of AKAP1 in regulation of mitochondrial biogenesis, dynamics, bioenergetics, and mitophagy. (**A**) AKAP1 influences mitochondrial biogenesis through the activation of PGC-1α, leading to increased production of mitochondrial proteins such as TFAM to promote mitochondrial biogenesis. Along with AKAP1, Ca^2+^ and AMP/ATP activity also promote mitochondrial biogenesis. (**B**,**C**) AKAP1 contributes to mitochondrial dynamics and bioenergetics. Specifically, it inhibits mitochondrial fission activity by preventing the activation of DRP1 through phosphorylation of the DRP1 S637 site. On the other hand, AKAP1 promotes mitochondrial fusion activity, which is mediated by OPA1 and MFN1/2. AKAP1 also promotes ATP synthesis by regulating OXPHOS complexes, resulting in balanced mitochondrial bioenergetics. (**D**) AKAP1 influences mitophagy mediated by the PINK1/Parkin pathway, where PINK1 accumulates in damaged mitochondria and recruits Parkin to degrade mitochondria through ubiquitination, which then promotes the formation of autophagosomes and eliminates the damaged mitochondria.

**Figure 3 cells-12-01539-f003:**
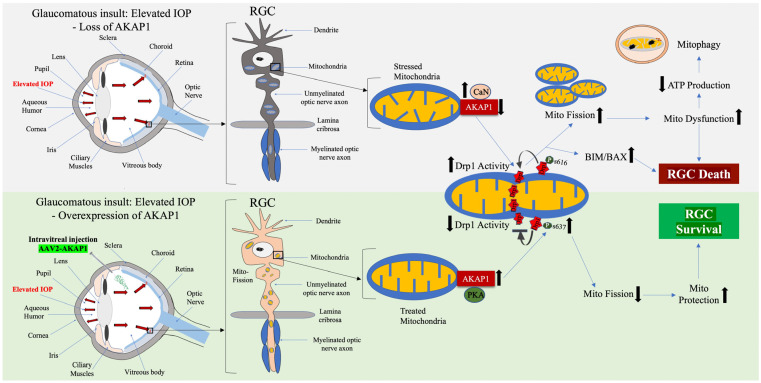
Neuroprotective effect of AKAP1 in glaucomatous RGCs. The upper panel diagrams the effect of glaucomatous insults such as elevated IOP in RGCs. Glaucomatous insults cause mitochondrial stress in RGCs, resulting in the loss of AKAP1, which triggers excessive mitochondrial fission through activation of DRP1 and the subsequent proapoptotic pathway BIM/BAX. This exacerbates mitochondrial dysfunction, ultimately leading to RGC death and a vision defect. The lower panel diagrams how restoring sustained AKAP1 expression in RGCs by intravitreal delivery of AAV2-AKAP1 protects RGCs by increasing phosphorylation of DRP1 S637, which promotes visual function in glaucomatous neurodegeneration.

## Data Availability

Not applicable.

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
