# Peer review of "Role of A-Kinase Anchoring Protein 1 in Retinal Ganglion Cells: Neurodegeneration and Neuroprotection"

_cells, 2023, doi:10.3390/cells12111539_

Round 1

Reviewer 1 Report

The review article by Bastola et al. entitled “Role of A-Kinase Anchoring Protein 1 in Retinal Ganglion Cells: Neurodegeneration and Neuroprotection” focuses on the role of AKAP1 in the maintenance of mitochondrial homeostasis in retinal ganglion cells (RGCs) via protein-protein interactions and intercellular signal transduction cascades that are involved in mitochondria dynamics, bioenergetics and mitophagy. Impairment of mitochondrial dynamics is associated with mitochondrial bioenergetics dysfunction, leading to neuronal loss in many neurodegenerative diseases, including glaucoma. Similar mechanisms were reported not only for glaucomatous neurodegeneration but also for other CNS neurodegenerative diseases (e.g. Alzheimer’s disease).

According to their findings and those of others, the authors suggest that amplified AKAP1 expression has the therapeutic potential to preserve mitochondrial homeostasis and promote RGC survival against glaucomatous insults such as elevated IOP and oxidative stress. They conclude by proposing further studies to help “identify the role of AKAP1 as a potential therapeutic target and open new innovative strategies for mitochondria dynamics-related neuroprotective intervention in glaucoma and other optic neuropathies”.

 This is an excellent review article and I recommend its publication in Cells. It is very well written and gives the reader everything he/she needs to know in order to understand the role of AKAP1 in mitochondria and the mechanisms involved (known till the present) that are involved in the neurodegeneration and neuroprotection of neurodegenerative diseases such as glaucoma.   

A few minor revisions are stated below.

1.      References: The reference list has many typos (e.g [16] Dickey, A.S. and S.J.J.o.N. Strack) and the presentation of the authors is not consistent (e.g. [27]. Schiattarella, G.G., et al.,). Please check typos and present all authors in all references.

2.       Line 384, ……… and bioenergetics, and further testing of the functional ...

Author Response

The authors thank the editor and the reviewers for thoughtful comments. We have summarized the suggestions and our responses in the rebuttal letter. The revisions in the text of the manuscript have been indicated in red color.
Reviewer #1:
  1. References: The reference list has many typos (e.g [16] Dickey, A.S. and S.J.J.o.N. Strack) and the presentation of the authors is not consistent (e.g. [27]. Schiattarella, G.G., et al.,). Please check typos and present all authors in all references.
Response: We have revised the references and used MDPI (Endnote) format as suggested.

  1. Line 384, ……… and bioenergetics, and further testing of the functional ... 
Response: We have revised the text as suggested (pg 9 and ln 396).

Reviewer 2 Report

The manuscript by Bastola et al. on the role of AKAP1 in RG cells: Neurodegeneration and Neuroprotection is an interesting topic for the review. Unfortunately, very little data are available on this topic. The authors managed to elaborate this review with concepts of mitochondrial dynamics and bioenergetics. However, I strongly encourage the authors to stick to main concept of AKAP1 in Neurodegeneration and protection instead of describing the basics of cAMP signaling (Page 3-lines 93-108), mitochondria dynamics (page 4-lines 140-151), bioenergetics (page 5, lines 178-187), mitophagy (page 6, 216-225), AD (page 8, 328-344). The above-mentioned texts are basics related to cAMP signaling, mitochondria or neural degeneration and not related to AKAP1 in RG cells or ND. Also, there are plenty of repetitions in the texts. The authors must avoid repeating the same or similar concepts.

Other comments:

1.     Abstract- First sentence has repetition and must be rephrased

2.     In the introduction, the authors can highlight the roles and importance of RG cells. General concepts of AKAPs in neuron functions can be mentioned.

3.     Figure 1. The authors must verify whether mouse Akap1 binds PP1

4.     Ref 18 is repeated as Ref 87. 

5.     Fig 2- lines 132-133. This line is a repetition

6.     Lines 164-165 is a repetition

7.     Lines 195-199 is a repetition

8.     Lines 302-311 about ref 18. Please check for repetition

9.     Lines 328-344 has nothing to do with PKA/AKAP/RGC.

10.  Spell check Fig 3- AAV2-AKAP1

11.  Fig 3 can be better described in the texts.

12.  Summary: Please offer future directions on this research topic and potential therapeutic options.

Author Response

The authors thank the editor and the reviewers for thoughtful comments. We have summarized the suggestions and our responses in the rebuttal letter. The revisions in the text of the manuscript have been indicated in red color.

Reviewer # 2:

  1. I strongly encourage the authors to stick to main concept of AKAP1 in neurodegeneration and protection instead of describing the basics of cAMP signaling (Page 3-lines 93-108), mitochondria dynamics (page 4-lines 140-151), bioenergetics (page 5, lines 178-187), mitophagy (page 6, 216-225), AD (page 8, 328-344). The above-mentioned texts are basics related to cAMP signaling, mitochondria or neural degeneration and not related to AKAP1 in RG cells or ND. Also, there are plenty of repetitions in the texts. The authors must avoid repeating the same or similar concepts.

Responses: This is a good point. We appreciate the reviewer’s a thoughtful comment regarding the description of the basic information provided on the topics (cAMP signaling, mitochondria dynamics, bioenergetics, mitophagy). We have strived to balance providing essential information and emphasizing the main concept of AKAP1. By including only the essential details to establish context for our discussion, we are confident that this approach will help reader’s understanding of the importance of our discussion and the potential therapeutic implications of AKAP1 in neurodegeneration diseases. Thus, we have revised the texts as follows:

cAMP signaling (Page 3-lines 93-108) to:

“cAMP, a ubiquitous second messenger, regulates growth, differentiation, and cell survival and

contributes to signal transduction downstream of growth factors, hormones, and neurotransmitters [27]. It is synthesized from the catalytic cyclization of ATP by adenylyl cyclases (ACs) [28] and acts through three types of effector proteins: PKA, exchange proteins activated by cAMP (EPACs), and cyclic nucleotide-gated ion channels (CNGs/HCNs) [29]. PKA is the major sensor of cAMP and could selectively phosphorylate substrates in response to different stimuli. The localization of PKA is guided by AKAPs that target it to specific substrates [30].” (pg 3 and lns 99-106)

Basic of mitochondria dynamics (page 4-lines 140-151) to:

“Mitochondria undergo four dynamic processes: fusion, fission, motility, and mi-tophagy. Mitochondrial fusion involves the joining of distinct mitochondria to form new mitochondria, while fission involves the division of a mitochondrion into separate mi-tochondria. Mitochondria are transported to specific sites within the cell by the micro-tubule network, a process called mitochondrial motility. Nonfunctional mitochondria are selectively eliminated through the mitophagy process, which involves machinery like that involved in autophagy [46]. Proper balance among these dynamic processes is critical for cellular homeostasis, and alteration in any of these processes has been linked to various pathological conditions. Multilayered regulatory mechanisms and signaling molecules ensure the proper tuning of these dynamic processes [47,48]. (pg 5 and lns 168-177)

Basic of bioenergetics (page 5, lines 178-187) to:

“Mitochondria provide energy to mammalian cells through the mitochondrial oxidative phosphorylation (OXPHOS) system. This system consists of five protein complexes and two electron carriers [59]. PKA regulates mitochondrial respiration by phosphorylating substrates like NDUFS4 and cytochrome c oxidase (COX) [60]. PKA-mediated phosphorylation of COX promotes allosteric ATP inhibition, while dephosphorylation by Ca2+ enhances respiration and ATP production [61,62]. It is unclear whether PKA acts at the IMM through holoenzymes or AKAP anchoring.” (pg 5 and lns 202-208)

Basic of mitophagy (page 6, 216-225) to:

“Mitochondrial dysfunction would be harmful to cells, but autophagy and mitophagy processes maintain energy metabolism in case of such dysfunction [68]. Mitophagy is the conserved cellular process that removes damaged mitochondria to preserve healthy ones [69] and can occur selectively or non-selectively [70]. In yeast, mitophagy is selective [69,71], whereas non-selective mitophagy happens during nutrient deprivation, when mitochondria are present in autophagosomes alongside other cellular components [72,73].” (pg 6 and lns 238-243)

Basic of AD (page 8, 328-344) to:

“AD is a prevalent neurodegenerative disease, potentially linked to mitochondrial dysfunction [98,106-110]. Mechanisms underlying mitochondrial dysfunction in AD include abnormal mitochondrial dynamics, deficits in mitochondrial trafficking and distribution, impaired mitochondrial biogenesis, abnormal ER-mitochondrial interaction, and impaired mitophagy. Mitochondrial dysfunction is an early and prominent feature of AD, suggesting a critical role of mitochondria in the early stage of AD pathogenesis [111,112]. Importantly, these pathological phenotypes of mitochondrial dysfunction in the brain of AD are like those in the glaucomatous retina, but the potential links and mechanisms underlying mitochondrial dysfunction in the retina of AD pathogenesis are not well understood.” (pg 8 and lns 345-354)

  1. Abstract- First sentence has repetition and must be rephrased

Response: We agree and have revised the text as “A-Kinase anchoring protein 1 (AKAP1) is a multifunctional mitochondrial scaffold protein that regulates mitochondrial dynamics, bioenergetics, and calcium homeostasis by anchoring several proteins, including protein kinase A, to the outer mitochondrial membrane.” (pg 1 and lns 18-20)

  1. In the introduction, the authors can highlight the roles and importance of RG cells. General concepts of AKAPs in neuron functions can be mentioned.

Response: We have added the roles and importance of RGCs cells as below:

“1.3. AKAP1 and Glaucomatous Retinal Ganglion cells

Retinal ganglion cells (RGCs) are solely responsible for transmitting visual information from the retina of the eye to the visual cortex of the brain. In humans, approximately 50% of RGC axons cross over to the opposite side at the optic chiasm, and most of these axons end up in a specific area of the brain called the lateral geniculate nucleus. However, in rodents, almost all RGC axons cross over at the optic chiasm, and some of axons form synapses in the superior colliculus [25-28]. RGC somas are located in the innermost layer of the retina, called the ganglion cell layer, and its dendrites ramify in the inner plexiform layer, where they receive signals from bipolar cell axons and amacrine cell processes. After processing this information, RGCs transmit it to the brain through their axons, which travel along the nerve fiber layer and optic nerve [29]. There are several types of RGCs, each with different properties and functions. Some RGCs are sensitive to light intensity and color, while others respond to movement or changes in the visual field. RGCs also vary in spatial resolution, with some cells able to detect fine details and others more suited to detecting larger objects [30]. In addition to their role in vision, RGCs also play an important role in other physiological processes, such as regulating circadian rhythms and modulating pupil size [31]. In glaucoma, RGCs and their axon degeneration are the main characteristics, resulting in vison loss. Until now, there are several risk factors in glaucoma pathophysiology, and these include elevated intraocular pressure (IOP) [32,33], oxidative stress [34], autoimmune response [35,36], glial cell activation [37,38], mitochondrial dysfunction [3], excitotoxicity [39] and aging [40]. However, the precise pathophysiological mechanisms underlying RGC soma and its axon degeneration still need to be elucidated. Evidence from our group demonstrated that mitochondrial dysfunction plays a critical role in optic nerve axon degeneration and progressive RGC death in glaucomatous neurodegeneration [3,41,42]. Since AKAP1 protects neuronal cells and promotes neurite outgrowth in the central nervous system [18], our recent studies indicate that glaucomatous insults such as elevated IOP and oxidative stress trigger AKAP1 deficiency in RGCs [43] and increasing AKAP1 expression promotes RGC survival against oxidative stress [3]. Thus, these findings suggest that AKAP1 may play an important role in mitochondrial protection in RGCs against glaucomatous neurodegeneration.” (pg 3 and lns 120-135; pg 4 and lns 136-149)

We have also added the general concepts of AKAPs in neuronal functions as below:

“Recent evidence indicates that AKAP-mediated signaling is critical for several fundamental neurological processes and suggests that understanding the function of AKAPs in nervous system disorders may develop novel therapeutic approaches [21]. AKAPs are involved in a wide range of neuronal functions, including neurotransmitter release [22], synaptic plasticity [23], neuronal excitability [24], ion channel function, and gene expression [25,26]. They are also crucial for adequately developing and maintaining neuronal structures such as dendrites and axons [16].” (pg 3 and lns 91-97)

  1. Figure 1. The authors must verify whether mouse Akap1 binds PP1

Response: The figure 1 was prepared based on the reference to [1] [Marin, Wenwen, 2020], which showed the binding of PP1 with mouse AKAP1. To confirm this, we have checked the specific genomic information of AKAP1 regarding to gene annotations, transcript variants and protein-coding in mouse from NCBI Gene (https://www.ncbi.nlm.nih.gov/gene/?term=akap1), which mentioned that “enables protein phosphatase binding”. However, we are not able to confirm about the specific aa sequence for the binding. Thus, we have corrected the figure as below.

  1. Ref 18 is repeated as Ref 87. 

Response: We have deleted the reference.

  1. Fig 2- lines 132-133. This line is a repetition.

Response: We revised the sentence as “AKAP1 contributes to mitochondrial dynamics and bioenergetics. Specifically, it inhibits mitochondrial fission activity by preventing the activation of DRP1 through phosphorylation of the DRP1 S637 site. On the other hand, AKAP1 promotes mitochondrial fusion activity, which is mediated by OPA1 and MFN1/2.” (pg 4 and lns 160-162)

  1. Lines 164-165 is a repetition.

Response: We have deleted the text “Previous studies demonstrated that PKA-mediated phosphorylation of DRP1 S637 164 prevented mitochondrial fission activity [33, 34]”.

  1. Lines 195-199 is a repetition.

Response: We have revised the line as " Flippo and coworker demonstrated that AKAP1 plays a critical role in maintaining the activity of respiratory complex II, an essential component of OXPHOS that generates ATP in mitochondria. In addition, deletion of AKAP1 dysregulates OXPHOS complex II, increases superoxide production, and impairs Ca2+ homeostasis in hippocampal neurons in response to glutamate excitotoxicity. These results suggest that AKAP1 is a key regulator of mitochondrial bioenergetics [2].” (pg 5 and lns 215-220; pg 6 and ln 221)

  1. Lines 302-311 about ref 18. Please check for repetition.

Response: We have revised the text as “Using a transient middle cerebral artery occlusion model of focal ischemia, Flippo and coworkers reported that AKAP1-deficient male mice were more vulnerable to focal brain ischemia and had smaller mitochondria yet a greater number of mitochondria and endoplasmic reticulum (ER) connections [18].” (pg 7 and lns 319-322)

  1. Lines 328-344 has nothing to do with PKA/AKAP/RGC.

Response: The preceding lines elucidated the broad contextual framework of the relationship between mitochondrial dysfunction and AD. Our argument posits that the neuroprotective function of AKAP1 can be demonstrated through its capacity to prevent mitochondrial dysfunction in neurodegenerative diseases such as AD. However, we agree and have revised the text as below:

“AD is a prevalent neurodegenerative disease, potentially linked to mitochondrial dysfunction [3-8]. Mechanisms underlying mitochondrial dysfunction in AD include abnormal mitochondrial dynamics, deficits in mitochondrial trafficking and distribution, impaired mitochondrial biogenesis, abnormal ER-mitochondrial interaction, and impaired mitophagy. Mitochondrial dysfunction is an early and prominent feature of AD, suggesting a critical role of mitochondria in the early stage of AD pathogenesis [9,10]. Importantly, these pathological phenotypes of mitochondrial dysfunction in the brain of AD are like those in the glaucomatous retina, but the potential links and mechanisms underlying mitochondrial dysfunction in the retina of AD pathogenesis are not well understood.” (pg 8 and lns 345-354)

  1. Spell check Fig 3- AAV2-AKAP1

Response: We have corrected the word in Figure 3.

  1. Fig 3 can be better described in the texts.

Response: The content of figure 3 was described in our manuscript and we present it as a graphical abstract of our current review. We have added some points to the text in the section 4. AKAP1 in Neuroprotection and described as “Thus, we proposed that sustained AKAP1 expression can inhibit CaN activation and promote DRP1 phosphorylation at the S637 site. This, in turn, can prevent mitochondrial fission or induce mitochondrial elongation and preserve mitochondrial function, leading to RGC survival (Figure 3).” (pg 8 and lns 373-376)

  1. Summary: Please offer future directions on this research topic and potential therapeutic options.

Response: We have added the text as “Since the amplification of AKAP1 gene expression by an in vitro transduction of AAV system is neuroprotective in cultured RGCs against oxidative stress [11], we will further validate the protective effect of AKAP1 on mitochondrial dynamics and bioenergetics in RGCs and their axons in experimental models of glaucoma using AAV-mediated gene therapeutic approaches.” (pg 10 and lns 413-418)

Round 2

Reviewer 2 Report

The authors mostly made the required changes.